# Bulk Operator Reconstruction in Topological Tensor Network and Generalized Free Fields

**DOI:** 10.3390/e25111543

**Published:** 2023-11-15

**Authors:** Xiangdong Zeng, Ling-Yan Hung

**Affiliations:** 1State Key Laboratory of Surface Physics, Fudan University, Shanghai 200433, China; 2Department of Physics and Center for Field Theory and Particle Physics, Fudan University, Shanghai 200433, China; 3Institute for Nanoelectronic Devices and Quantum Computing, Fudan University, Shanghai 200433, China; 4Yau Mathematical Sciences Center, Tsinghua University, Beijing 100084, China; 5Yanqi Lake Beijing Institute of Mathematical Sciences and Applications (BIMSA), Huairou District, Beijing 101408, China

**Keywords:** bulk operator reconstruction, tensor network, topological field theory

## Abstract

In this paper, we study operator reconstruction in a class of holographic tensor networks describing renormalization group flows studied in arXiv:2210.12127. We study examples of 2D bulk holographic tensor networks constructed from Dijkgraaf–Witten theories and find that for both the $Z_{n}$ group and the $S_{3}$ group, the number of bulk operators behaving like a generalized free field in the bulk scales as the order of the group. We also generalize our study to 3D bulks and find the same scaling for $Z_{n}$ theories. However, there is no generalized free field when the bulk comes from more generic fusion categories such as the Fibonacci model.

## 1. Introduction

An important feature of the AdS/CFT correspondence [1] in the large *N* limit is the emergence of free fields propagating in the AdS bulk, allowing a semi-classical description of the bulk theory and computation of non-trivial quantities in the CFT via the bulk. By free fields, we mean that the correlation functions of these operators are computed approximately by Wick contraction. Generically, single trace operators in the CFT become approximately free Gaussian fields in the semi-classical dual bulk theory. This feature allows performance of computations of correlation functions via Witten diagrams in the AdS side [2,3]. The emergence of generalized free fields in holographic CFT has also been discussed extensively (see, for example, [4,5,6], and very recently [7] and references therein).

Along a different vein, inspired by the Ryu–Takayanagi formula [8], it has been proposed that the tensor network is the appropriate framework to construct the linear map between the CFT degrees of freedom and the AdS ones [9]. The graphical representation of the appropriate tensor network looks like a discrete version of the AdS bulk. It describes a kind of a coarse-graining process for boundary degrees of freedom, which also reside at the (asymptotic boundary) of the tensor network.

The bulk operators act on the auxiliary legs of the tensor network, and the CFT operators act on the legs located at the asymptotic boundary of the network. Clearly, the tensor network is providing a linear map between these operators [10,11].

Because the tensor network is local, i.e., it is decomposed into a product of tensors that only contract with some number of neighboring tensors (the precise number depends on the dimension of the bulk space), the bulk boundary map can be read off via “operator pushing” [10] (examples are also discussed extensively in [12,13]).

This leads us to a natural question: what kind of tensor networks most resemble the bulk-boundary map encountered in AdS/CFT where the bulk is describable by a weakly coupled bulk theory where the bulk fields are almost free? This question was addressed in [13]. Bulk operators act on internal legs in the bulk. Under operator pushing, its action on the tensor is equivalent to that of another set of operators acting on some other legs. This is illustrated in Figure 1. In such a reconstruction, a choice has to be made over which of the legs are in-coming and which are out-going. There is a canonical choice when the tensor network is essentially a holographic tensor network performing coarse-graining, where operators acting on degrees of freedom after coarse-graining should be related to operators acting on the pre-coarse-grained degrees of freedom. In this case, it was shown [13] that a generalized free field should be such that when the operators are pushed across a tensor, they can be decomposed as a sum of “simple operators”. To be precise, if the tensor network is made up of a network of coarse-graining tensors $M_{j_{1}\cdots j_{H}}^{i_{1}\cdots i_{K}}$ that take $V^{K}\to V^{H}$ (where K>H, $i_{i},j_{i}\in V$, and V is a *d*-dimensional vector space), then operator pushing corresponds to
(1)$$O\left(V^{H}\right)\cdot M=M\cdot O\left(V^{K}\right).$$
An almost free bulk operator acting on one of the legs among $V^{H}$, i.e.,
(2)$$O\left(V^{H}\right)=I_{1}⊗\cdots ⊗I_{i-1}⊗X_{i}⊗I_{i+1}⊗\cdots ⊗I_{H},$$
satisfies Equation (Equation 1) such that
(3)$$O\left(V^{K}\right)=\sum_{i=1}^{K}\alpha_{i}\;\left(I_{1}⊗\cdots ⊗I_{i-1}⊗X_{i}⊗I_{i+1}⊗\cdots ⊗I_{K}\right)$$
for some constants $\left\{\alpha_{i}\right\}$. When this is satisfied, the reconstruction of the bulk operator $O(l_{B})$ acting on a bulk leg $l_{B}$ in terms of boundary operators takes the form of
(4)$$O\left(l_{B}\right)=\sum_{b}K^{I}\left(l_{b},l_{B}\right)O^{I}\left(l_{b}\right),$$
where $\left\{O^{I}\right\}$ is a complete set of basis operators acting on each leg $l_{b}$ at the boundary, and $K^{I}\left(l_{b},l_{B}\right)$ is the bulk-boundary kernel which can be expressed in terms of $\alpha_{i}$ in Equation (Equation 3). This expression has the same form as the HKLL kernel constructed from bulk boundary propagators [14,15], and it can be shown that correlation functions of bulk operators can behave like generalized free fields, i.e., that correlation functions can be approximated by Wick contraction, and that other connected components can be expanded perturbatively by Feynman-like diagrams [13,16]. We note that since *M* is a rectangular matrix, the reconstruction is not unique. However, a generalized free field is one where Equation (Equation 3) can be satisfied at all.

In this paper, we explore families of coarse-graining tensor networks that follow from topological theories, introduced and discussed in [17,18]. They are interesting because they are key to recovering families of CFTs, and the coarse-graining or RG tensors carry resemblances to the AdS bulk which is checked numerically at least in low dimensions. It is thus interesting to study operator pushing in these RG tensor networks and explore when generalized free fields might emerge.

We begin our analysis with RG operators constructed from 1 + 1D topological field theory. In particular, we focus on the untwisted version of Dijkgraaf–Witten theories with gauge group *G*. It is found that for $G=Z_{n}$ there are *n* generalized free fields. We also study the simplest non-abelian theory with $G=S_{3}$.

This study is generalized to RG operators constructed from 2 + 1D topological field theory. We study both the Dijkgraaf–Witten type lattice gauge theories and the simple examples of Turaev–Viro-type theories. In the case of $Z_{n}$ lattice gauge theories, it is found that the number of generalized free fields is given by *n*. We also compute examples of operator reconstruction in the stereotypical example of topological orders in 2 + 1 dimensions, namely the Fibonacci model. In this case, we find no generalized free operator at all.

## 2. Operator Pushing in 1 + 1D

We first consider the 1 + 1D untwisted Dijkgraaf–Witten theory characterized by group *G*. An RG operator can be constructed from the topological theory [17,18], which takes the form of a tree network, as illustrated in Figure 2. Each vertex is three-valent, and for the untwisted version of the Dijkgraaf–Witten theory, each three-valent vertex resides in a three-index tensor $M_{g_{3}}^{g_{1},g_{2}}$ that takes the following form [18]: 
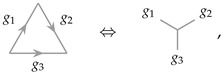
(5)
where $g_{1}$, $g_{2}$ and $g_{3}$ are elements of *G*. Tensor *M* imposes the group product or fusion rule of *G*, such that it is non-vanishing only for $G\left(g_{1},g_{2}\right):=g_{1}\times g_{2}=g_{3}$, i.e.,
(6)$$M_{k}^{ij}=\delta_{G(i,j),k}.$$

The tensor network has a 1D boundary (see Figure 2). When inserting operator *B* in the bulk, its action is equivalent to some other operators at the boundary. Finding the boundary operator that recreates the action of a given bulk operator is the problem of bulk operator reconstruction. Since the tensor network is local, we can reconstruct the action of the bulk operator by studying the reconstruction of the bulk operator across one constituent tensor in the RG tensor network. Specifically, reconstruction or operator pushing across one constituent tensor amounts to finding operator *A* for a given operator *B*, such that

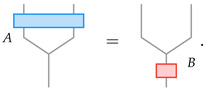
(7)
In the case where bulk operator *B* is a “generalized free field”, the reconstruction by *A* should be expressible in a “simple” form as discussed above (i.e., *A* is a generic linear combination of $I⊗\tilde{A}$ and $\tilde{A}⊗I$). To ensure that the operator behaves like a generalized free field, we require that $\tilde{A}$ falls back into the set of operators {B} that admits simple reconstruction.

The general reconstruction equation for one constituent tensor is given by
(8)$$A_{\left(ij\right),\left(i^{\prime }j^{\prime }\right)}M_{k}^{i^{\prime }j^{\prime }}=M_{k^{\prime }}^{ij}B_{k^{\prime }k}.$$
In the following, we consider solving Equation (Equation 8) for generic operator *B* and also identifying the set of generalized free fields from it. We note that the collection of *B* whose reconstruction *A* that is “simple” forms a complete basis of generalized free fields in the tensor network.

The vector space residing on each leg of a tensor would generically be of dimension |G|. We can construct a basis for the operators acting on each leg using the generalized Pauli matrices [19]. For a finite group *G* with |G|=n and elements labeled by 0,1,…,n−1, we can construct basis states for the |G| dimensional vector space at each leg given by
(9)$$|i\rangle =\left(\begin{array}{c} 0\\ \vdots \\ 1_{i-\mathrm{th}}\\ \vdots \\ 0 \end{array}\right),\;i\in \left\{0,1,\ldots ,n-1\right\}.$$
In such basis, the generalized Pauli matrices are generated by *shift matrix X* and *clock matrix Z* as follows:(10)$$X=\left(\begin{array}{ccccc} 0 & 1 & 0 & \cdots & 0\\ 0 & 0 & 1 & \cdots & 0\\ \vdots & \vdots & \vdots & \ddots & \vdots \\ 0 & 0 & 0 & \cdots & 1\\ 1 & 0 & 0 & \cdots & 0 \end{array}\right),\;Z=\left(\begin{array}{ccccc} 1 & 0 & 0 & \cdots & 0\\ 0 & \omega & 0 & \cdots & 0\\ \vdots & \vdots & \vdots & \ddots & \vdots \\ 0 & 0 & 0 & \cdots & \omega^{n-1} \end{array}\right),$$
where $\omega =e^{2\pi i/n}$ is the *n*th root of unity. Then, the generalized Pauli matrices are
(11)$$\sigma_{\mu }:=\sigma_{ns+t}=X^{t}Z^{s},$$
where s=⌊μ/n⌋, t=μmodn and $\mu \in \{0,1,\ldots ,n^{2}-1\}$. Now, Equation (Equation 8) becomes
(12)$$A_{\left(ij\right),\left(i^{\prime }j^{\prime }\right)}\delta_{G(i^{\prime },j^{\prime }),k}=\delta_{G\left(i,j\right),k^{\prime }}B_{k^{\prime }k}=\delta_{G\left(i,j\right),k^{\prime }}{\left(\sigma_{\mu }\right)}_{k^{\prime }k}={\left(\sigma_{\mu }\right)}_{G(i,j),k}.$$
Thus, once G(i,j) is specified, we can solve the above equation for each $\sigma_{\mu }$ via standard linear algebra methods. A solution is given by
(13)$$A_{\left(ij\right),\left(i^{\prime }j^{\prime }\right)}^{\left(\mu \right)}=\left\{\begin{array}{cc} {\left(\sigma_{\mu }\right)}_{G\left(i,j\right),j^{\prime }}, & i^{\prime }=0;\\ 0. & i^{\prime }\neq 0. \end{array}\right.$$
As expected, since *M* is a rectangular matrix, the solution above is only determined up to the addition of any linear combination of the collection of homogeneous solutions satisfying $A_{\left(ij\right),\left(i^{\prime }j^{\prime }\right)}\delta_{G(i^{\prime },j^{\prime }),k}=0$.

We look for the subset of bulk operators such that there exists *A* that is a local/simple operator, and that it admits non-trivial action only on one leg, and acts trivially on the rest, i.e.,
(14)$$A_{\left(ij\right),\left(i^{\prime }j^{\prime }\right)}={\tilde{A}}_{ii^{\prime }}\delta_{jj^{\prime }}\;\mathrm{or}\;\delta_{ii^{\prime }}{\tilde{A}}_{jj^{\prime }}.$$
Then,
(15)$${\tilde{A}}_{ii^{\prime }}\delta_{G(i^{\prime },j),k}={\left(\sigma_{\mu }\right)}_{G(i,j),k}\;\mathrm{or}\;{\tilde{A}}_{jj^{\prime }}\delta_{G(i,j^{\prime }),k}={\left(\sigma_{\mu }\right)}_{G(i,j),k},\;\forall j\in \left\{0,1,\ldots ,n-1\right\}.$$
$\tilde{A}$ can be solved, too. The necessary and sufficient condition that the above equation has a solution is that for each column of ${\left[{\left(\sigma_{\mu }\right)}_{G(i,j),k}\right]}_{s}^{T}$, the rank of the augmented matrix (i.e., appending this column vector to the matrix) satisfies
(16)$$\mathrm{rank}\left[{\left(\delta_{G(i^{\prime },j),k}\right)}^{T}|{\left({\left(\sigma_{\mu }\right)}_{G(i,j),k}\right)}_{s}^{T}\right]=\mathrm{rank}\left[{\left(\delta_{G(i^{\prime },j),k}\right)}^{T}\right]=\mathrm{rank}\left(\delta_{G(i^{\prime },j),k}\right)=n.$$
Practically, we can take coefficients $\left\{\alpha_{\mu }\right\}$ such that
(17)$${\tilde{A}}_{ii^{\prime }}\delta_{G(i^{\prime },j),k}-\sum_{\mu }\alpha_{\mu }{\left(\sigma_{\mu }\right)}_{G(i,j),k}=0\;\mathrm{or}\;{\tilde{A}}_{jj^{\prime }}\delta_{G(i,j^{\prime }),k}-\sum_{\mu }\alpha_{\mu }{\left(\sigma_{\mu }\right)}_{G(i,j),k}=0,$$
which is a set of homogeneous equations with respect to ${\tilde{A}}_{ii^{\prime }}$ (or ${\tilde{A}}_{jj^{\prime }}$) and $\alpha_{\mu }$. The null space of the coefficient matrix
(18)$$\tilde{M}=\left(\begin{array}{cccccccccc} \delta_{G(0,0),0} & \cdots & \delta_{G(n-1,0),0} & \cdots & 0 & \cdots & 0 & {\left(\sigma_{0}\right)}_{G(0,0),0} & \cdots & {\left(\sigma_{n^{2}-1}\right)}_{G(0,0),0}\\ \vdots & \ddots & \vdots & \ddots & \vdots & \ddots & \vdots & \vdots & \ddots & \vdots \\ \delta_{G(0,0),n-1} & \cdots & \delta_{G(n-1,0),n-1} & \cdots & 0 & \cdots & 0 & {\left(\sigma_{0}\right)}_{G(0,0),n-1} & \cdots & {\left(\sigma_{n^{2}-1}\right)}_{G(0,0),n-1}\\ \vdots & \ddots & \vdots & \ddots & \vdots & \ddots & \vdots & \vdots & \ddots & \vdots \\ \delta_{G(0,n-1),n-1} & \cdots & \delta_{G(n-1,n-1),n-1} & \cdots & 0 & \cdots & 0 & {\left(\sigma_{0}\right)}_{G(0,n-1),n-1} & \cdots & {\left(\sigma_{n^{2}-1}\right)}_{G(0,n-1),n-1}\\ \vdots & \ddots & \vdots & \ddots & \vdots & \vdots & \vdots & \vdots & \ddots & \vdots \\ 0 & \cdots & 0 & \cdots & \delta_{G(0,0),0} & \cdots & \delta_{G(n-1,0),0} & {\left(\sigma_{0}\right)}_{G(0,0),0} & \cdots & {\left(\sigma_{n^{2}-1}\right)}_{G(0,0),0}\\ \vdots & \ddots & \vdots & \ddots & \vdots & \ddots & \vdots & \vdots & \ddots & \vdots \\ 0 & \cdots & 0 & \cdots & \delta_{G(0,n-1),n-1} & \cdots & \delta_{G(n-1,n-1),n-1} & {\left(\sigma_{0}\right)}_{G(0,n-1),n-1} & \cdots & {\left(\sigma_{n^{2}-1}\right)}_{G(0,n-1),n-1} \end{array}\right)$$
of Equation (Equation 17) (in ${\tilde{A}}_{ii^{\prime }}$ case, for example) then gives $B=\sum_{\mu }\alpha_{\mu }\sigma_{\mu }$ and the corresponding *A* operator.

In the following, we present some concrete results for special classes of theories.

### 2.1. Invitation: $Z_{2}$

We begin with the example of $Z_{2}$ group, where the tensor unit is given by
(19)$$M=\delta_{G(i,j),k}=\delta_{(i+j)\;\mathrm{mod}\;2,k}=\left(\begin{array}{cc} 1 & 0\\ 0 & 1\\ 0 & 1\\ 1 & 0 \end{array}\right).$$
It is a rank-2 matrix, and the null space of $M^{T}$ is spanned by
(20)$$\left\{v^{\left(p\right)}\right\}=\left\{\left(\begin{array}{c} 1\\ 0\\ 0\\ -1 \end{array}\right),\;\left(\begin{array}{c} 0\\ 1\\ -1\\ 0 \end{array}\right)\right\}.$$
The general solution $A^{*}$ is given by the linear combination of $v^{\left(p\right)}$:(21)$$A^{*}=\left(\begin{array}{cccc} \beta_{0,0} & \beta_{0,1} & -\beta_{0,1} & -\beta_{0,0}\\ \beta_{1,0} & \beta_{1,1} & -\beta_{1,1} & -\beta_{1,0}\\ \beta_{2,0} & \beta_{2,1} & -\beta_{2,1} & -\beta_{2,0}\\ \beta_{3,0} & \beta_{3,1} & -\beta_{3,1} & -\beta_{3,0} \end{array}\right),$$
where $\beta_{ij}$ are arbitrary constants. The full solutions are then $A^{*}$ plus the specific solution part in Equation (Equation 13):
$$\begin{array}{cc} B=\sigma_{0}=I & \Rightarrow A=A^{*}+\left(\begin{array}{cccc} 1 & 0 & 0 & 0\\ 0 & 1 & 0 & 0\\ 0 & 1 & 0 & 0\\ 1 & 0 & 0 & 0 \end{array}\right), \end{array}$$
$$\begin{array}{cc} B=\sigma_{1}=\sigma_{x} & \Rightarrow A=A^{*}+\left(\begin{array}{cccc} 0 & 1 & 0 & 0\\ 1 & 0 & 0 & 0\\ 1 & 0 & 0 & 0\\ 0 & 1 & 0 & 0 \end{array}\right), \end{array}$$
$$\begin{array}{cc} B=\sigma_{2}=\sigma_{z} & \Rightarrow A=A^{*}+\left(\begin{array}{cccc} 1 & 0 & 0 & 0\\ 0 & -1 & 0 & 0\\ 0 & -1 & 0 & 0\\ 1 & 0 & 0 & 0 \end{array}\right), \end{array}$$
(22)$$\begin{array}{cc} B=\sigma_{3}=-i\sigma_{y} & \Rightarrow A=A^{*}+\left(\begin{array}{cccc} 0 & -1 & 0 & 0\\ 1 & 0 & 0 & 0\\ 1 & 0 & 0 & 0\\ 0 & -1 & 0 & 0 \end{array}\right). \end{array}$$
We can see that only $\sigma_{x}$ inserted in bulk corresponds to a simple form
(23)$$A=I⊗\sigma_{x}=\left(\begin{array}{cccc} 0 & 1 & 0 & 0\\ 1 & 0 & 0 & 0\\ 0 & 0 & 0 & 1\\ 0 & 0 & 1 & 0 \end{array}\right)\;\mathrm{or}\;\sigma_{x}⊗I=\left(\begin{array}{cccc} 0 & 0 & 1 & 0\\ 0 & 0 & 0 & 1\\ 1 & 0 & 0 & 0\\ 0 & 1 & 0 & 0 \end{array}\right)$$
at the boundary. Therefore, there is only one non-trivial generalized free field in $Z_{2}$, other than the identity operator which certainly can be written in the simple form.

### 2.2. Abelian Case: $Z_{n}$

Having understood the structure of the $Z_{2}$ case, here, we generalize the computation to $Z_{n}$. The fusion rules are given by modular arithmetic:(24)G(i,j)=(i+j)modn.
Then, $\delta_{G(i,j),k}=\delta_{(i+j)\;\mathrm{mod}\;n,k}$ is a rank-*n* matrix, whose (transposed) null space is spanned by $n^{2}-n$ vectors $v^{\left(p\right)}$ such that
(25)$$v_{q}^{\left(p\right)}=\delta_{(-p-\lfloor p/n\rfloor -2)\;\mathrm{mod}\;n,q}-\delta_{n^{2}-p-1,q}$$
where $p\in \{0,1,\ldots ,n^{2}-n-1\}$, $q\in \{0,1,\ldots ,n^{2}-1\}$. The general solution $A^{*}$ is given by the linear combination of $v^{\left(p\right)}$:(26)$$A^{*}=\left(\begin{array}{c} \beta_{0,0}v^{\left(0\right)}+\cdots +\beta_{0,n^{2}-n-1}v^{\left(n^{2}-n-1\right)}\\ \vdots \\ \beta_{n^{2}-1,0}v^{\left(0\right)}+\cdots +\beta_{n^{2}-1,n^{2}-n-1}v^{\left(n^{2}-n-1\right)} \end{array}\right),$$
where $\beta_{ij}$ are arbitrary constants. The specific solution part is
(27)$$A_{\left(ij\right),\left(0j^{\prime }\right)}^{\left(\mu \right)}={\left(\sigma_{\mu }\right)}_{\left(i+j\right)\;\mathrm{mod}\;n,j^{\prime }},$$
as we already mentioned in Equation (Equation 13).

To solve for bulk operators with local/simple reconstruction, it can be seen that the coefficient matrix $\tilde{M}$ of Equation (Equation 17) now has dimension $n^{3}\times 2n^{2}$ and rank $2n^{2}-n$. The solution is
(28)$${\tilde{A}}_{ii^{\prime }}^{\left(k\right)}={\left(\sigma_{k}\right)}_{ii^{\prime }},\;\alpha_{\mu }^{\left(k\right)}=\delta_{k\mu },$$
where k∈{0,1,…,n−1}. Thus, it means that the generalized free bulk operator given by $B=\sigma_{k}$ corresponds to a simple operator
(29)$$A=I⊗\sigma_{k}\;\mathrm{or}\;\sigma_{k}⊗I.$$
Since $\tilde{A}=\sigma_{k}$ falls in the set of *B*, this operator pushing procedure can be further iterated.

When considering the full holographic network that is a tree with many layers of the constituent tensors studied above (e.g., with a total number of layers *L*, see Figure 2), we can still find a solution by iteratively using the above method. When $B=\sigma_{k}$ where k∈{0,1,…,n−1}, since
(30)$$A_{1}=\sigma_{k}⊗I\;\mathrm{or}\;I⊗\sigma_{k},$$
the boundary operator at level *L* is still simple and can be written as
(31)$$A_{L}=I^{⊗L-l}⊗\sigma_{k}⊗I^{⊗l-1},\;l=0,1,\ldots ,L.$$
Any other generalized free bulk operators can be constructed from linear combinations of these *B* operators. To conclude, there are exactly n−1 non-trivial generalized free bulk operators emerging from the tensor network. In the large *n* limit, there is a large number of generalized free fields scaling with the rank of the group, which is very different from a usual holographic theory.

### 2.3. Non-Abelian Case: $S_{3}$

The group multiplication table for $S_{3}$ is

$g_{0}$$g_{1}$$g_{2}$$g_{3}$$g_{4}$$g_{5}$$g_{0}$012345$g_{1}$103254$g_{2}$240513$g_{3}$351402$g_{4}$425031$g_{5}$534120
We denote $G\left(i,j\right)=g_{i}g_{j}$, e.g.,
(32)$$G\left(1,2\right)=g_{1}g_{2}=g_{3}:=3,\;G\left(2,1\right)=g_{2}g_{1}=g_{4}:=4.$$
Now, the dimension and rank of matrix $\delta_{G(i,j),k}$ is 36×6 and 6, respectively. The null space of $\delta_{G(i,j),k}^{T}$ is thus spanned by 30 vectors $v^{\left(p\right)}$ such that
(33)$$v^{\left(p\right)}=\left(\begin{array}{c} {\tilde{v}}^{\left(p\right)}\\ {\hat{v}}^{\left(p\right)} \end{array}\right),\;p\in \left\{0,1,\ldots ,29\right\},$$
where ${\tilde{v}}^{\left(p\right)}$ are length-6 vectors:(34)$$\begin{array}{cc} {\tilde{v}}^{\left(1\right)}={\tilde{v}}^{\left(8\right)}={\tilde{v}}^{\left(16\right)}={\tilde{v}}^{\left(21\right)}={\tilde{v}}^{\left(29\right)} & ={\left(1,0,0,0,0,0\right)}^{T},\\ {\tilde{v}}^{\left(0\right)}={\tilde{v}}^{\left(10\right)}={\tilde{v}}^{\left(14\right)}={\tilde{v}}^{\left(23\right)}={\tilde{v}}^{\left(27\right)} & ={\left(0,1,0,0,0,0\right)}^{T},\\ {\tilde{v}}^{\left(3\right)}={\tilde{v}}^{\left(6\right)}={\tilde{v}}^{\left(17\right)}={\tilde{v}}^{\left(19\right)}={\tilde{v}}^{\left(28\right)} & ={\left(0,0,1,0,0,0\right)}^{T},\\ {\tilde{v}}^{\left(2\right)}={\tilde{v}}^{\left(11\right)}={\tilde{v}}^{\left(12\right)}={\tilde{v}}^{\left(22\right)}={\tilde{v}}^{\left(25\right)} & ={\left(0,0,0,1,0,0\right)}^{T},\\ {\tilde{v}}^{\left(5\right)}={\tilde{v}}^{\left(7\right)}={\tilde{v}}^{\left(15\right)}={\tilde{v}}^{\left(18\right)}={\tilde{v}}^{\left(26\right)} & ={\left(0,0,0,0,1,0\right)}^{T},\\ {\tilde{v}}^{\left(4\right)}={\tilde{v}}^{\left(9\right)}={\tilde{v}}^{\left(13\right)}={\tilde{v}}^{\left(20\right)}={\tilde{v}}^{\left(24\right)} & ={\left(0,0,0,0,0,1\right)}^{T}, \end{array}$$
and ${\hat{v}}^{\left(p\right)}$ are length-30 vectors:(35)$${\hat{v}}_{q}^{\left(p\right)}=\delta_{pq},\;p,q\in \left\{0,1,\ldots ,29\right\}.$$
The general solution is then given by the linear combination of these $v^{\left(p\right)}$ plus the specific part $A_{\left(ij\right),\left(0j^{\prime }\right)}^{\left(\mu \right)}={\left(\sigma_{\mu }\right)}_{G\left(i,j\right),j^{\prime }}$ as in Equation (Equation 13).

For the solution leading to simple forms in the reconstruction, the coefficients in Equation (Equation 17) form a 216×72 matrix $\tilde{M}$, whose rank is 66. Hence, there are six solutions for $A={\tilde{A}}_{L}⊗I$, which are given by
$$\begin{array}{cccc} {\tilde{A}}_{L}^{\left(0\right)}=B_{L}^{\left(0\right)} & =\left(\begin{array}{cccccc} 1 & 0 & 0 & 0 & 0 & 0\\ 0 & 1 & 0 & 0 & 0 & 0\\ 0 & 0 & 1 & 0 & 0 & 0\\ 0 & 0 & 0 & 1 & 0 & 0\\ 0 & 0 & 0 & 0 & 1 & 0\\ 0 & 0 & 0 & 0 & 0 & 1 \end{array}\right), & {\tilde{A}}_{L}^{\left(1\right)}=B_{L}^{\left(1\right)} & =\left(\begin{array}{cccccc} 0 & 1 & 1 & 1 & 1 & 1\\ 1 & 0 & 1 & 1 & 1 & 1\\ 1 & 1 & 0 & 1 & 1 & 1\\ 1 & 1 & 1 & 0 & 1 & 1\\ 1 & 1 & 1 & 1 & 0 & 1\\ 1 & 1 & 1 & 1 & 1 & 0 \end{array}\right), \end{array}$$
$$\begin{array}{cccc} {\tilde{A}}_{L}^{\left(2\right)}=B_{L}^{\left(2\right)} & =\left(\begin{array}{cccccc} 0 & -1 & 1 & 1 & 1 & 1\\ -1 & 0 & 1 & 1 & 1 & 1\\ 1 & 1 & 0 & -1 & 1 & 1\\ 1 & 1 & -1 & 0 & 1 & 1\\ 1 & 1 & 1 & 1 & 0 & -1\\ 1 & 1 & 1 & 1 & -1 & 0 \end{array}\right), & {\tilde{A}}_{L}^{\left(3\right)}=B_{L}^{\left(3\right)} & =\left(\begin{array}{cccccc} 0 & 0 & 1 & 0 & 0 & 0\\ 0 & 0 & 0 & 0 & 1 & 0\\ 1 & 0 & 0 & 0 & 0 & 0\\ 0 & 0 & 0 & 0 & 0 & 1\\ 0 & 1 & 0 & 0 & 0 & 0\\ 0 & 0 & 0 & 1 & 0 & 0 \end{array}\right), \end{array}$$
(36)$$\begin{array}{cccc} {\tilde{A}}_{L}^{\left(4\right)}=B_{L}^{\left(4\right)} & =\left(\begin{array}{cccccc} 0 & 0 & \frac{1}{2} & -\omega & \omega & -1\\ 0 & 0 & \omega & -1 & \frac{1}{2} & -\omega \\ \frac{1}{2} & -\omega & 0 & 0 & -1 & \omega \\ \omega & -1 & 0 & 0 & -\omega & \frac{1}{2}\\ -\omega & \frac{1}{2} & -1 & \omega & 0 & 0\\ -1 & \omega & -\omega & \frac{1}{2} & 0 & 0 \end{array}\right), & {\tilde{A}}_{L}^{\left(5\right)}=B_{L}^{\left(5\right)} & =\left(\begin{array}{cccccc} 0 & 0 & \frac{5}{2} & \eta & \bar{\eta } & 1\\ 0 & 0 & \bar{\eta } & 1 & \frac{5}{2} & \eta \\ \frac{5}{2} & \eta & 0 & 0 & 1 & \bar{\eta }\\ \bar{\eta } & 1 & 0 & 0 & \eta & \frac{5}{2}\\ \eta & \frac{5}{2} & 1 & \bar{\eta } & 0 & 0\\ 1 & \bar{\eta } & \eta & \frac{5}{2} & 0 & 0 \end{array}\right), \end{array}$$
where $\omega =e^{\pi i/3}$ and $\eta =-\frac{1+3\sqrt{3}i}{2}$. Since $S_{3}$ is non-abelian, we need to separately solve for $A=I⊗{\tilde{A}}_{R}$, where the solutions are
$$\begin{array}{cccc} {\tilde{A}}_{R}^{\left(0\right)}=B_{R}^{\left(0\right)} & =\left(\begin{array}{cccccc} 1 & 0 & 0 & 0 & 0 & 0\\ 0 & 1 & 0 & 0 & 0 & 0\\ 0 & 0 & 1 & 0 & 0 & 0\\ 0 & 0 & 0 & 1 & 0 & 0\\ 0 & 0 & 0 & 0 & 1 & 0\\ 0 & 0 & 0 & 0 & 0 & 1 \end{array}\right), & {\tilde{A}}_{R}^{\left(1\right)}=B_{R}^{\left(1\right)} & =\left(\begin{array}{cccccc} 0 & 1 & 1 & 1 & 1 & 1\\ 1 & 0 & 1 & 1 & 1 & 1\\ 1 & 1 & 0 & 1 & 1 & 1\\ 1 & 1 & 1 & 0 & 1 & 1\\ 1 & 1 & 1 & 1 & 0 & 1\\ 1 & 1 & 1 & 1 & 1 & 0 \end{array}\right), \end{array}$$
$$\begin{array}{cccc} {\tilde{A}}_{R}^{\left(2\right)}=B_{R}^{\left(2\right)} & =\left(\begin{array}{cccccc} 0 & 1 & 1 & 1 & 1 & -2\\ 1 & 0 & 1 & 1 & -2 & 1\\ 1 & 1 & 0 & -2 & 1 & 1\\ 1 & 1 & -2 & 0 & 1 & 1\\ 1 & -2 & 1 & 1 & 0 & 1\\ -2 & 1 & 1 & 1 & 1 & 0 \end{array}\right), & {\tilde{A}}_{R}^{\left(3\right)}=B_{R}^{\left(3\right)} & =\left(\begin{array}{cccccc} 0 & 0 & 0 & 1 & 0 & 0\\ 0 & 0 & 1 & 0 & 0 & 0\\ 0 & 0 & 0 & 0 & 0 & 1\\ 0 & 0 & 0 & 0 & 1 & 0\\ 1 & 0 & 0 & 0 & 0 & 0\\ 0 & 1 & 0 & 0 & 0 & 0 \end{array}\right), \end{array}$$
(37)$$\begin{array}{cccc} {\tilde{A}}_{R}^{\left(4\right)}=B_{R}^{\left(4\right)} & =\left(\begin{array}{cccccc} 0 & -\omega & \omega^{2} & 1 & 1 & -\frac{1}{2}\\ -\omega & 0 & 1 & \omega^{2} & -\frac{1}{2} & 1\\ \omega^{2} & 1 & 0 & -\frac{1}{2} & -\omega & 1\\ 1 & \omega^{2} & -\frac{1}{2} & 0 & 1 & -\omega \\ 1 & -\frac{1}{2} & -\omega & 1 & 0 & \omega^{2}\\ -\frac{1}{2} & 1 & 1 & -\omega & \omega^{2} & 0 \end{array}\right), & {\tilde{A}}_{R}^{\left(5\right)}=B_{R}^{\left(5\right)} & =\left(\begin{array}{cccccc} 0 & \xi & \bar{\xi } & 1 & 1 & 1\\ \xi & 0 & 1 & \bar{\xi } & 1 & 1\\ \bar{\xi } & 1 & 0 & 1 & \xi & 1\\ 1 & \bar{\xi } & 1 & 0 & 1 & \xi \\ 1 & 1 & \xi & 1 & 0 & \bar{\xi }\\ 1 & 1 & 1 & \xi & \bar{\xi } & 0 \end{array}\right), \end{array}$$
where $\xi =-2+\sqrt{3}i$. It can be seen that all the $\tilde{A}$ falls in the set of *B*, so the operator pushing procedure can be iterated to high levels as well. Perhaps surprisingly, even for non-Abelian groups, the number of generalized free fields continues to scale with the rank of the group, as in the $Z_{n}$ case.

We also note that two solutions in each part coincide, i.e., ${\tilde{A}}_{L}^{\left(0\right)}={\tilde{A}}_{R}^{\left(0\right)}$ and ${\tilde{A}}_{L}^{\left(1\right)}={\tilde{A}}_{R}^{\left(1\right)}$. It indicates the abelian $Z_{2}$ subgroup of $S_{3}$.

## 3. Operator Pushing in 2 + 1d

The construction and analysis in the previous section have a natural generalization in one higher dimension. Families of 3D (or 2 + 1D) holographic tensor networks can be constructed from Levin–Wen string net models. The holographic tensor networks constructed from Levin–Wen models were discussed in [18]. The unit tensor constituting the tensor network is represented as a tetrahedron drawn in Figure 3a.

In Figure 3b,c, the RG operator takes edges i,j,m,n to *a* (with b,c,d,e as spectators untouched). Therefore, the problem of operator reconstruction can be understood as reconstructing the bulk operator *B* acting on leg *a* by operators *A* acting on legs i,j,m,n. We also find the set of operators that are generalized free operators.

Each tetrahedron has six edges which are labeled by the objects in a fusion category characterizing the topological field theory. For a given labeled tetrahedron, the value of an *F*-symbol is assigned depending on the labels on the six edges. It is non-vanishing if there exists a fusion channel for the three edges of every triangle to fuse to the trivial identity object. Because of this constraint, it is more convenient to think of each allowed configuration of a triangle as our fundamental degrees of freedom (in the bulk) and to consider operators acting on the triangle that transform them between allowed configurations. The problem of operator pushing across the unit tensor can be formulated as pushing the bulk operator action on the triangle basis ${▵}_{abc}$ or ${▵}_{ade}$.

Since Figure 3c is symmetric, we only need to consider one tetrahedron such as abcinm and the other can be simply obtained by flipping all legs. Here, we use the following convention: lowercase letters such as i,j∈{0,1,…,n−1} are reserved for edge labels, where *n* is the number of objects in the fusion category; uppercase I,J∈{0,1,…,N−1} are triangle (or face, or fusion channel) labels, and *N* is the number of admissible configurations on a triangle. The tetrahedron can then be labeled as

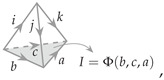
(38)
where I=Φ(b,c,a) is the triangle label, and function Φ can be determined by fusion rules.

The constraint equation in 2 + 1D is similar to the 1 + 1D case Equations (Equation 8) and (Equation 12):(39)$$A_{\left(ijk\right),\left(i^{\prime }j^{\prime }k^{\prime }\right)}M_{(i^{\prime }j^{\prime }k^{\prime }),I}=M_{\left(ijk\right),I^{\prime }}B_{I^{\prime }I}=M_{\left(ijk\right),I^{\prime }}{\left(\sigma_{\mu }\right)}_{I^{\prime }I},$$
where $\mu \in \{0,1,\ldots ,N^{2}-1\}$ labels the generalized Pauli matrices in the triangle basis. Now, *M* is determined by
(40)$$M_{(ijk),I}=M_{\left(ijk),\Phi (b,c,a\right)}=\sqrt{d_{j}d_{k}d_{b}d_{c}}\;{\left[F_{c}^{jkb}\right]}_{ia},$$
where ${\left[F_{c}^{jkb}\right]}_{ia}$ is the *F*-symbol and $d_{i}$ is quantum dimension of object *i* in the fusion category.

Similarly, to check whether *A* can be written in a “simple” form, we look for *A* of the form, say,
(41)$$A_{\left(ijk\right),\left(i^{\prime }j^{\prime }k^{\prime }\right)}={\tilde{A}}_{ii^{\prime }}\delta_{jj^{\prime }}\delta_{kk^{\prime }},$$
such that
(42)$${\tilde{A}}_{ii^{\prime }}M_{(i^{\prime }jk),I}=M_{\left(ijk\right),I^{\prime }}{\left(\sigma_{\mu }\right)}_{I^{\prime }I},\;\forall j,k\in \left\{0,1,\ldots ,n-1\right\}.$$
The condition for simple form reconstruction is thus
(43)$$\mathrm{rank}\left[{\left(M_{(i^{\prime }jk),I}\right)}^{T}|{\left(M_{\left(ijk\right),I^{\prime }}{\left(\sigma_{\mu }\right)}_{I^{\prime }I}\right)}_{s}^{T}\right]=\mathrm{rank}\left(M_{(i^{\prime }jk),I}\right).$$
We can also change it to
(44)$${\tilde{A}}_{ii^{\prime }}M_{(i^{\prime }jk),I}-\sum_{\mu }\alpha_{\mu }M_{\left(ijk\right),I^{\prime }}{\left(\sigma_{\mu }\right)}_{I^{\prime }I}=0$$
to explicitly solve *B*. Taking indices *i*, *j* or *i*, *k* to be identical mappings leads to other simple form solutions as well.

### 3.1. $Z_{n}$ Case

We again look into the simple example where the fusion category is the abelian group $Z_{n}$. The objects that label the edges of the tetrahedron are again taken from group elements of $Z_{n}$. We recall that in each admissible triangle, two of the edges must fuse to the third, and fusion here again means a group product of $Z_{n}$, which was introduced in Equation (Equation 24). The number of admissible triangles is thus given by $N=n^{2}$, i.e., the triangles labels lie in $\left\{0,1,\ldots ,n^{2}-1\right\}$. Taking $Z_{2}$ as an example, the admissible triangles (it is also customary to use the dual graph of the triangle to highlight the fusion relation between the edges, which we adopt below) are listed in the following. We also assign a label to each admissible triangle, from 0 to 3.

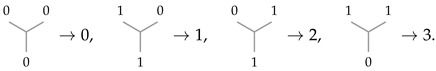
(45)
The above triangle labeling rule for generic *n* can be stated as
(46)I=nb+c=n[(i+j)modn]+[(i+k)modn],
where i,j are two of the edges of the triangle (say, the two top edges when adopting notation as in Equation (45)). We consider the class of trivial $Z_{n}$ Dijkgraaf–Witten models, where all *F*-symbols equal unity when fusion constraints at each triangle are satisfied (for generic $Z_{n}$ group where the group elements are generically not self-dual, we have to put in an orientation on each edge (see Figure 3c). This can be achieved by numbering the vertices and attaching an arrow to each edge pointing from the vertex with a smaller number to the other with a larger label. The result is independent of such labeling). We have
(47)$$M_{(ijk),I}=\delta_{n[(i+j)\;\mathrm{mod}\;n]+[(i+k)\;\mathrm{mod}\;n],I}.$$
Therefore, the constraint equation becomes
$$A_{\left(ijk\right),\left(i^{\prime }j^{\prime }k^{\prime }\right)}\delta_{n\left[\left(i^{\prime }+j^{\prime }\right)\;\mathrm{mod}\;n\right]+\left[\left(i^{\prime }+k^{\prime }\right)\;\mathrm{mod}\;n\right],I}={\left(\sigma_{\mu }\right)}_{n[(i+j)\;\mathrm{mod}\;n]+[(i+k)\;\mathrm{mod}\;n],I}.$$
Now, $M^{T}$ is a rank-$n^{2}$ matrix, whose null space is spanned by $n^{3}-n^{2}$ vectors $v^{\left(p\right)}$ such that
(48)$$v_{q}^{\left(p\right)}=\delta_{n\left[\left(-\left\lfloor p/n\right\rfloor -\left\lfloor p/n^{2}\right\rfloor -2\right)\;\mathrm{mod}\;n\right]+\left[\left(-\left\lfloor p/n^{2}\right\rfloor -2\right)\;\mathrm{mod}\;n\right],q}-\delta_{n^{3}-p-1,q},$$
where $p\in \{0,1,\ldots ,n^{3}-n^{2}-1\}$, $q\in \{0,1,\ldots ,n^{3}-1\}$ and the specific solution part is
(49)$$A_{\left(ijk\right),\left(0j^{\prime }k^{\prime }\right)}^{\left(\mu \right)}={\left(\sigma_{\mu }\right)}_{n\left[\left(i+j\right)\;\mathrm{mod}\;n\right]+\left[\left(i+k\right)\;\mathrm{mod}\;n\right],nj^{\prime }+k^{\prime }}.$$

For the simple operator case, we check Equation (Equation 44), whose coefficients form an $n^{5}\times \left(n^{4}+n^{2}\right)$ matrix with rank $n^{4}+n^{2}-n$, so its null space is spanned by *n* vectors. In general *n*, it is difficult to solve Equation (Equation 44) and we only consider small *n* here. For n=2, we have
(50)$$\begin{array}{cc} B^{\left(0\right)} & =\left(\begin{array}{cccc} 1 & 0 & 0 & 0\\ 0 & 1 & 0 & 0\\ 0 & 0 & 1 & 0\\ 0 & 0 & 0 & 1 \end{array}\right)=\sigma_{0}^{\left(2\right)}⊗\sigma_{0}^{\left(2\right)},\\ B^{\left(1\right)} & =\left(\begin{array}{cccc} 0 & 0 & 0 & 1\\ 0 & 0 & 1 & 0\\ 0 & 1 & 0 & 0\\ 1 & 0 & 0 & 0 \end{array}\right)=\sigma_{1}^{\left(2\right)}⊗\sigma_{1}^{\left(2\right)} \end{array}$$
and
(51)$${\tilde{A}}^{\left(0\right)}=\sigma_{0}^{\left(2\right)},\;{\tilde{A}}^{\left(1\right)}=\sigma_{1}^{\left(2\right)}.$$
For n=3, the solutions are
(52)$$B^{\left(0\right)}=\sigma_{0}^{\left(3\right)}⊗\sigma_{0}^{\left(3\right)},\;B^{\left(1\right)}=\sigma_{1}^{\left(3\right)}⊗\sigma_{1}^{\left(3\right)},\;B^{\left(2\right)}=\sigma_{2}^{\left(3\right)}⊗\sigma_{2}^{\left(3\right)}$$
and
(53)$${\tilde{A}}^{\left(0\right)}=\sigma_{0}^{\left(3\right)},\;{\tilde{A}}^{\left(1\right)}=\sigma_{1}^{\left(3\right)},\;{\tilde{A}}^{\left(2\right)}=\sigma_{2}^{\left(3\right)}.$$
Here, we use the superscript of σ to denote its size for clarity.

Other simple form solutions can be calculated in the same way. When $A_{\left(ijk\right),\left(i^{\prime }j^{\prime }k^{\prime }\right)}={\tilde{A}}_{jj^{\prime }}\delta_{ii^{\prime }}\delta_{kk^{\prime }}$, we have
(54)$$B^{\left(0\right)}=\sigma_{0}^{\left(2\right)}⊗\sigma_{0}^{\left(2\right)},\;B^{\left(1\right)}=\sigma_{1}^{\left(2\right)}⊗\sigma_{0}^{\left(2\right)}$$
for $Z_{2}$ and
(55)$$B^{\left(0\right)}=\sigma_{0}^{\left(3\right)}⊗\sigma_{0}^{\left(3\right)},\;B^{\left(1\right)}=\sigma_{1}^{\left(3\right)}⊗\sigma_{0}^{\left(3\right)},\;B^{\left(2\right)}=\sigma_{2}^{\left(3\right)}⊗\sigma_{0}^{\left(3\right)}$$
for $Z_{3}$; when $A_{\left(ijk\right),\left(i^{\prime }j^{\prime }k^{\prime }\right)}={\tilde{A}}_{kk^{\prime }}\delta_{ii^{\prime }}\delta_{jj^{\prime }}$, we have
(56)$$B^{\left(0\right)}=\sigma_{0}^{\left(2\right)}⊗\sigma_{0}^{\left(2\right)},\;B^{\left(1\right)}=\sigma_{0}^{\left(2\right)}⊗\sigma_{1}^{\left(2\right)}$$
for $Z_{2}$ and
(57)$$B^{\left(0\right)}=\sigma_{0}^{\left(3\right)}⊗\sigma_{0}^{\left(3\right)},\;B^{\left(1\right)}=\sigma_{0}^{\left(3\right)}⊗\sigma_{1}^{\left(3\right)},\;B^{\left(2\right)}=\sigma_{0}^{\left(3\right)}⊗\sigma_{2}^{\left(3\right)}$$
for $Z_{3}$. The corresponding $\tilde{A}$ are the same as in Equations (Equation 51) and (Equation 53).

In Figure 3c, *B* operators in the two tetrahedra act on triangles ${▵}_{abc}$ and ${▵}_{ade}$. In the above equations, we see that *B* can be decomposed into small σ that act on the edges. For the next iteration of operator pushing, where the bulk is now given by triangles ${▵}_{bim}$, ${▵}_{cin}$, ${▵}_{djm}$ and ${▵}_{ejn}$, it can be seen that the *A* operators acting on edge *i* and *j*, as well as the decomposed *B* operators on edge *b*, *c*, *d*, *e*, altogether offer the new *B* operator for these four triangles. Therefore, the operator pushing procedure can be reiterated at the next level.

### 3.2. Fibonacci Model

Another important example of topological order in the 2 + 1 dimension is the Fibonacci model. As a fusion category, there are two objects 1 and τ. They satisfy the following fusion rules:(58)1⊗1=1,1⊗τ=τ⊗1=τ,τ⊗τ=1⊕τ,
so there are five admissible triangles: 
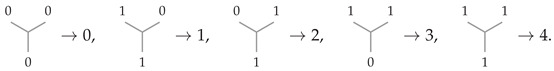
(59)
The non-vanishing *F* symbols are
(60)$$F_{\tau }^{\tau \tau \tau }=\left(\begin{array}{cc} \phi^{-1} & \phi^{-1/2}\\ \phi^{-1/2} & -\phi^{-1} \end{array}\right)$$
where $\phi =(1+\sqrt{5})/2$ denotes the golden ratio. Then,
(61)$$M_{(ijk),I}=\left(\begin{array}{ccccc} 1 & 0 & 0 & 0 & 0\\ 0 & 0 & \phi & 0 & 0\\ 0 & 0 & 0 & \phi & 0\\ 0 & \phi & 0 & 0 & \phi^{3/2}\\ 0 & \phi & 0 & 0 & 0\\ 0 & 0 & 0 & \phi & \phi^{3/2}\\ 0 & 0 & \phi & 0 & \phi^{3/2}\\ \phi & \phi^{3/2} & \phi^{3/2} & \phi^{3/2} & -\phi \end{array}\right).$$
We can see rank(M)=5 so the null space of $M^{T}$ is spanned by three vectors:(62)$$\left\{v^{\left(p\right)}\right\}=\left\{\left(\begin{array}{c} \phi^{1/2}\\ 1\\ 1\\ -\phi^{-1}\\ \phi \\ 0\\ 0\\ -\phi^{1/2} \end{array}\right),\;\left(\begin{array}{c} 0\\ 1\\ 0\\ 1\\ -1\\ 0\\ -1\\ 0 \end{array}\right),\;\left(\begin{array}{c} 0\\ 0\\ 1\\ 1\\ -1\\ -1\\ 0\\ 0 \end{array}\right)\right\}.$$
The specific solution part for each $\sigma_{\mu }$ can thus be calculated; see Appendix A.

For the simple form, we can see the coefficient matrix of Equation (Equation 44) is $\left(2^{3}\times 5\right)\times \left(2^{2}+5^{2}\right)=40\times 29$, but it has rank 28, so there is only one solution. However, since the trivial solution ($\tilde{A}$ and *B* are both identity operators) always exists, there does not exist a non-trivial generalized free field taking the form of
(63)$$A=\tilde{A}⊗I⊗I\;\mathrm{or}\;I⊗\tilde{A}⊗I\;\mathrm{or}\;I⊗I⊗\tilde{A}$$
in the Fibonacci model.

We should mention that although not presented here, the above method is in principle applicable to any fusion category such as Ising and $A_{k+1}$ models. However, the size of matrix *M* and $\tilde{M}$ grows quickly with the number of objects, so it is difficult to solve the constraint equation for very large systems.

## 4. Conclusions

In this paper, we explore the emergence of generalized free fields in classes of holographic tensor networks constructed from topological field theories. We consider both 1 + 1D and 2 + 1d networks and demonstrate, for example, in networks following from abelian Dijkgraaf–Witten theories in the trivial cohomological class, the number of generalized free fields scale with the rank of the group. Interestingly, for the simple case of a Fibonacci model in 2 + 1D constructed from the first non-abelian fusion category with the lowest rank, its RG network admits no generalized free field. Of course, to recover a holographic network that resembles a semi-classical bulk theory, it is expected that the spectrum has a large gap with only a sparse number of free fields as the degrees of freedom (i.e., central charge) approaches infinity. While we do not expect such a simple model to admit generalized free fields, it is interesting to find out under what circumstances they would admit some. In the case of RG operators that follow from Dijkgraaf–Witten lattice gauge theories, there are way too many generalized free fields and it is unlikely they are the right candidate for generating a holographic dual with a semi-classical limit. Generic fusion categories however could be plausible.

These models are not expected to recover a semi-classical bulk theory, although they are examples where the dual CFT can be constructed explicitly. It would be interesting to study more generic TQFT with other large “rank” limit to look for bulk networks with semi-classical approximations. These interesting problems are left for future investigations.

## Figures and Tables

**Figure 1 entropy-25-01543-f001:**
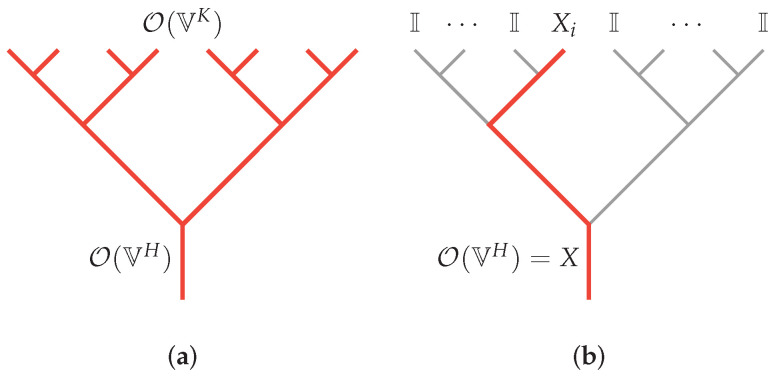
Operator pushing in a holographic tensor network that is performing coarse-graining. Here, we illustrate a tree-like holographic tensor network. (**a**) Bulk operator $O(V^{H})$ is reconstruced by boundary operator $O(V^{K})$, where we take H=1 for simplicity. (**b**) Bulk operator $O(V^{H})=X$ is reconstruced by a simple form boundary operator $O\left(V^{K}\right)=\sum_{i}\alpha_{i}\left(I_{1}⊗\cdots ⊗X_{i}⊗\cdots ⊗I_{K}\right)$ as in Equation (Equation 3). Hence, it corresponds to a generalized free field.

**Figure 2 entropy-25-01543-f002:**
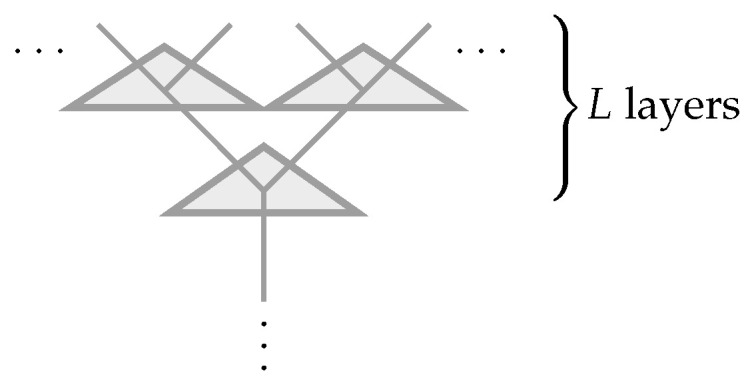
RG operator in 1 + 1D Dijkgraaf–Witten theory characterized by group *G*, which takes the form of a tree tensor network with 1D boundary and total number of layers *L*. The legs are elements of group *G*. Explicitly, for the untwisted version of Dijkgraaf–Witten theory, each of these three-valent vertices (or dual triangles) resides a tensor $M_{g_{3}}^{g_{1},g_{2}}=\delta_{G\left(g_{1},g_{2}\right),g_{3}}:=\delta_{g_{1}\times g_{2},g_{3}}$. Images from [18] (with modification).

**Figure 3 entropy-25-01543-f003:**
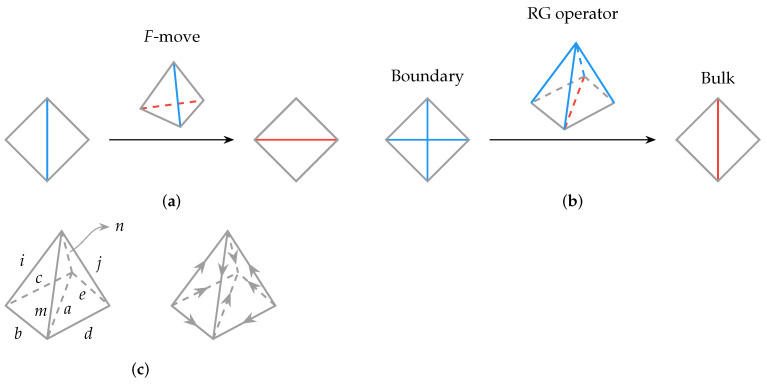
(**a**) A single tetrahedron is used as *F*-move that changes the surface triangulation. It is assigned the value of an *F*-symbol depending on the labels on the 6 edges. The blue and red lines should be summed over. (**b**) With two or more tetrahedra, we can construct the RG operator that maps the smaller triangles at the boundary onto larger triangles in the bulk. (**c**) Arrows on each edge indicate the direction of fusions. If all the objects are self-dual, then these arrows can be ignored.

## Data Availability

No new data were created or analyzed in this study. Data sharing is not applicable to this article.

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
