# Peer review of "Bulk Operator Reconstruction in Topological Tensor Network and Generalized Free Fields"

_entropy, 2023, doi:10.3390/e25111543_

Round 1
Reviewer 1 Report
Comments and Suggestions for Authors
This paper studies operator reconstruction in holographic tensor networks constructed from topological field theories. The authors focus on networks built from untwisted Dijkgraaf-Witten theories and the Fibonacci model, in both 1+1d and 2+1d. The key results are:
- For $Z_n$ theories, the number of bulk operators behaving like generalized free fields scales with the order n of the group.
- For the non-abelian $S_3$ group, there are still several generalized free fields, with the number scaling with the group size.
- In the Fibonacci model example, there are no non-trivial generalized free fields.
The paper is well-written and the calculations are clearly explained. Mapping out the emergence of generalized free fields in tensor networks is an interesting direction. This helps identify networks that can recover holographic theories with weakly coupled semi-classical gravity duals. I recommend publication after the following suggestions are considered:
(1) It would be helpful to provide a brief definition of "generalized free field" around line 54 or 88, to give readers some intuition for the criterion and in what sense the operators are "free".
(2) What is $\tilde{M}$ in line 120? This coefficient matrix seems undefined. Clarifying its relation to the RG tensor $M$ would be useful.
(3) Could the authors add some physical intuition for why only $\sigma^x$ (or $\sigma_k$) operators have a simple reconstruction? Is it likely related to the fact that they commute with the $Z_n$ symmetry and are symmetric operators? Is it general that simple operators must be symmetric in this framework?
(4) In 1+1d the RG tensor is constructed within a single simplex, but in 2+1d it uses multiple simplices. What goes wrong with defining the 2+1d RG tensor on a single F-move? Some comments on this difference would be interesting.
In summary, this is a valuable study of generalized free fields in tensor networks, providing insights into holographic emergent gravity. The results are novel and clearly presented. I recommend publication in Entropy after the minor points above are addressed.
Author Response
We thank the referees for their careful reading of our manuscript, and their encouragements and useful comments.
We would like to answer the questions that the referees have raised in the following.
(1) It would be helpful to provide a brief definition of "generalized free field" around line 54 or 88, to give readers some intuition for the criterion and in what sense the operators are "free".
Reply: "Generalized free field" refers to the fact that the correlation function of this operator approximately satisfies Wick's theorem and that non-trivial connected correlation functions can be computed using the analogues of Feynman diagrams, in which propagators meet at an interaction vertex, and that each interaction vertex involving 3-point or more is suppressed by a small coupling. This question is explored in our previous works arXiv:1703.05445 in the context of tensor networks, where an action is not explicitly available. It was observed that when an operator in the bulk of the tensor network acting on a subset of the legs are reconstructed from operators acting on another set of the legs (this choice set would be determined more clearly when the network takes the form of the RG operator, as it is in the current paper) and that the operator retains its form under such reconstruction, i.e. remain in the same form acting locally on the legs. When this is satisfied, it then becomes possible to construct the propagator of this operator and it would satisfy approximately the Wick's theorem as observed in arXiv:1703.05445. We have included the content of this paragraph into the introduction of the paper to make it clearer.
(2) What is $\tilde{M}$ in line 120? This coefficient matrix seems undefined. Clarifying its relation to the RG tensor $M$ would be useful.
Reply: We add the explicit form of $\tilde{M}$ for clarification.
(3) Could the authors add some physical intuition for why only $\sigma^x$ (or $\sigma_k$) operators have a simple reconstruction? Is it likely related to the fact that they commute with the $Z_n$ symmetry and are symmetric operators? Is it general that simple operators must be symmetric in this framework?
Reply: In the $Z_2$ theory this is understandable because operator pushing in the context of the $Z_2$ group corresponds to multiplying to the leg by a non-trivial $Z_2$ group element. When it is pushed across the network it corresponds to distributing the $Z_2$ action to the legs across, and so using the $Z_2$ group multiplication rule one sees that the non-trivial action can only be distributed onto the other side one by one democratically. This logic can be generalized to other group elements in the $Z_n$ group, where one can select the group action to be reproduced by distributing the group element onto exactly one other leg across the tensor which are intertwiners of the group.
(4) In 1+1d the RG tensor is constructed within a single simplex, but in 2+1d it uses multiple simplices. What goes wrong with defining the 2+1d RG tensor on a single F-move? Some comments on this difference would be interesting.
Reply: By RG operator, we mean a map that map N indices in boundary to n indices in bulk, where N > n. However, if using a single simplex (tetrahedron), than it's actually a 1 to 1 map with other 4 legs unchanged (Fig 3.a). When employing two tetrahedra, then the map is 4 to 1 which can be viewed as a valid RG operator (Fig 3.b). The secret is that these two tetrahedra share a common face (triangle [amn] in Fig 3.c) and the two F-moves give a common output leg [a]. This is also addressed in our previous paper arXiv:2210.12127, especially in Fig 6.
Reviewer 2 Report
Comments and Suggestions for Authors
The manuscript discusses examples of bulk-boundary maps in the context of holographic tensor network models. The authors specifically address topological field theory models (Dijkgraaf–Witten theories) in 1+1 and 2+1 dimensions, leading to Bulk reconstruction of generalized free field operators.
The main result in the paper is the scaling of the number of certain bulk operators (those behaving like generalized free fields) as the order of the group. The authors find that this scaling holds for different examples of Dijkgraaf–Witten theories, abelian and nonabelian. But it does not hold for the Fibonacci theory (a nonabelian 2+1 D theory).
This is not a very important result, but the paper provides explicit computations of the results and I believe that would be a valuable addition to the literature on this subject. Even though the results are simple, the analysis presented in the work is very illuminating and may be a starting point for further investigations. The authors also provide a nice discussion in the conclusions regarding their findings.
Therefore, I recommend the paper for publication in this journal.
Comments on the Quality of English Language
The paper is well written but there are many English grammar typos throughout the text and these should be corrected before publication.
Author Response
We thank the referees for their careful reading of our manuscript, and their encouragements and useful comments.
The paper is well written but there are many English grammar typos throughout the text and these should be corrected before publication.
We will correct these typos in the updated version.
Round 2
Reviewer 1 Report
Comments and Suggestions for Authors
The authors have resolved my comments. I recommend publication of this work.